# A simplified minimodel of visual cortical neurons

Fengtong Du [1] ✉, Miguel Angel Núñez-Ochoa [1], Marius Pachitariu [1,2] & Carsen Stringer [1,2] ✉

Artificial neural networks (ANNs) have been shown to predict neural responses in primary visual cortex (V1) better than classical models. However, this performance often comes at the expense of simplicity and interpretability. Here we introduce a new class of simplified ANN models that can predict over 70% of the response variance of V1 neurons. To achieve this high performance, we first recorded a new dataset of over 29,000 neurons responding to up to 65,000 natural image presentations in mouse V1. We found that ANN models required only two convolutional layers for good performance, with a relatively small first layer. We further found that we could make the second layer small without loss of performance, by fitting individual "minimodels" to each neuron. Similar simplifications applied for models of monkey V1 neurons. We show that the minimodels can be used to gain insight into how stimulus invariance arises in biological neurons.

Predictive models of neural activity have a long tradition in neuroscience. Such models have many uses, from making predictions of responses to new stimuli, to developing normative and prescriptive theories of neural coding, to making testable hypotheses about underlying mechanisms, etc[1,2]. Predictive models range from simple qualitative descriptions (i.e., V1 neurons are edge detectors), to complex mathematical functions with very many parameters and nonlinearities (i.e., deep convolutional neural networks). The performance of a model can be directly measured by its prediction accuracy on new stimuli; complex models typically excel at this. However, predictive power is not everything; simple models can often be more useful for understanding neural coding properties.

Simple models of V1 responses include simple/complex cell models[3], linear-nonlinear (LN) models[4,5], Gabor functions[6], and orientation tuning curves[7,8]. These models are simple and interpretable, yet they struggle to capture the complex feature selectivity observed in natural visual environments[9]. Complex models in contrast are predominantly artificial neural networks (ANNs) in various configurations, mostly deep convolutional neural networks[10] but more recently transformer models as well[11,12]. Initially developed and demonstrated as models of higher-order areas in the primate brain[13,14], ANN models have also been found to perform well in primate V1[15] and in mouse V1[16,17],

where they can predict almost twice as much variance on test images compared to linear and LN models. These results have challenged the traditional view of V1 neurons as simple edge detectors and filter banks. Simultaneously, other results in mice have shown that V1 neurons represent many other behavioral and cognitive variables in addition to representing stimuli[18,19], although these variables do not appear to be represented in primate V1[20]. Thus, a view of a complicated V1 is emerging, that is potentially different between mouse and monkey.

Here we aimed to directly test whether complex, many-stage neural networks are an appropriate model of V1 in both mouse and monkey. Our approach is to start with multi-layer neural networks, which are known to predict V1 responses well, and progressively remove parts from the model for as long as the performance stays the same. To enable these analyses, we also recorded a new large dataset of high-quality neural responses from tens of thousands of V1 neurons to tens of thousands of images[21], well beyond the dataset sizes previously employed in similar studies.

## Results

### Data and model setup

Using a two-photon microscope, we recorded neural activity from a total of 29,608 V1 neurons in six mice expressing jGCaMP8s[22] in

[1]HHMI Janelia Research Campus, Ashburn, VA, USA. [2]These authors jointly supervised this work: Marius Pachitariu, Carsen Stringer.
✉e-mail: fengtongd@janelia.hhmi.org; stringerc@janelia.hhmi.org

excitatory neurons (Supplementary Fig. 1). During the recordings, we presented 32,440–52,868 naturalistic images at a stimulus presentation rate of 7.5 Hz[22,23] (Fig. 1a, Supplementary Movie 1). A subset of 500 images were presented 10 times and were used as test images. Similar to previous studies, we restricted all analyses to neurons with reliable stimulus responses to the test images[15,24], which resulted in 14,504 neurons (Supplementary Fig. 2a, b). These neurons had Gabor-like linear receptive fields, similar to reported monkey receptive fields (Supplementary Fig. 3,[25]).

We fit various models to predict the responses of these neurons using the training images as input. We started from a neural network model with four convolutional layers, fit to a neural population, that is similar to previous work[24,26] (see "Methods"). The four convolutional layers were shared across neurons (the "core" of the model) and were followed by a neuron-specific readout step which pooled over the output of the last convolutional layer (Fig. 1b). The readout was parameterized as a rank-1 decomposition of weights across horizontal pixels ($\mathbf{w}_x$), vertical pixels ($\mathbf{w}_y$), and convolutional channels ($\mathbf{w}_c$), further simplifying the readout models used in previous work[27], and we constrained the $\mathbf{w}_x$ and $\mathbf{w}_y$ to be non-negative. We did not put any spatial constraints on $\mathbf{w}_x$ and $\mathbf{w}_y$.

To quantify the performance of the model, we used the responses to a separate set of 500 test images, repeated 10 times each. As a performance metric, we used the fraction of explainable variance explained (FEVE), which is the ratio of variance explained to total explainable variance (similar to previous studies[15,24], see Methods). The standard model reached 0.73 FEVE on the mouse data (Fig. 1c, d;

Supplementary Fig. 2), significantly outperforming a LN model (FEVE = 0.31). The performance was also higher than previously reported values of 0.44 FEVE from another dataset[24]. The increase in performance can likely be explained by the larger number of training images we were able to show ( ~30,000 vs ~5000) as well as the SNR increase obtained by jGCaMP8s recordings under good recording conditions with closed-loop eye correction (Fig. 1e and "Methods"). We also found that the model performance was not related to the single neuron response statistics: we fit a Gabor model to each neuron and did not find any relationship between FEVE and the Gabor parameters, including whether or not the cell was "complex" and its spatial frequency preference (Supplementary Fig. 4 and "Methods").

## Simplifying V1 models to two layers

To determine the simplest model that performs well, we first varied the number of convolutional layers. We found that the FEVE metric saturates at two convolutional layers (Fig. 1f, 0.61, 0.71, 0.73, 0.73 FEVE for one-, two-, three- and four-layer models averaged across mice). Previous high-performing models in contrast saturated more slowly (Fig. 1f, 0.37, 0.61, 0.67, 0.68 FEVE for one-, two-, three- and four-layer models). We found that the performance gains in our shallower models compared to the Sensorium model were primarily due to the larger pooling area in the readout weights ($\mathbf{w}_x$, $\mathbf{w}_y$) of our models (Supplementary Fig. 5). The $\mathbf{w}_x$ and $\mathbf{w}_y$ readout weights were spatially localized, i.e., the model pooled across the same local region multiple different inputs (Fig. 1g, Supplementary Fig. 6). The pooling

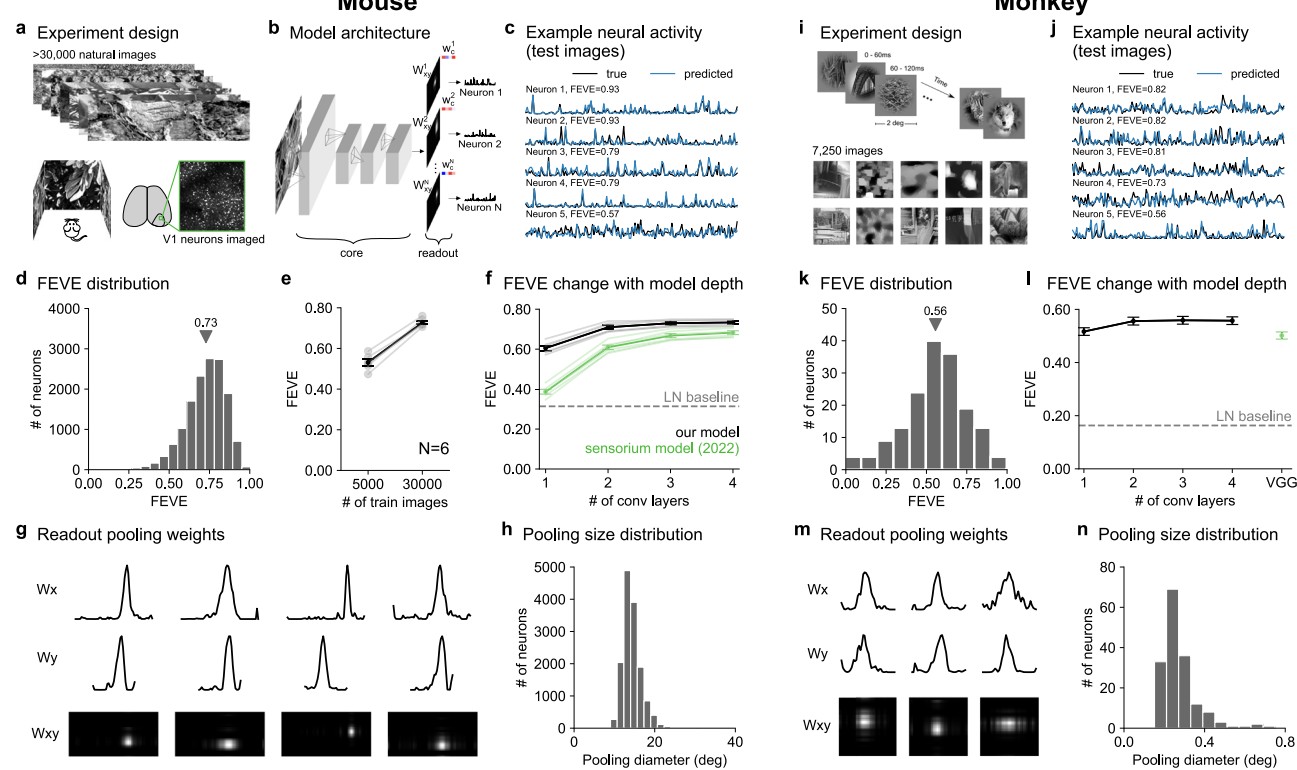

**Fig. 1 | Two-layer models of visual responses in mouse and monkey V1.**
**a** 32,440–52,868 natural images were shown to mice during two-photon calcium imaging recordings in V1. **b** Architecture of the prediction model including four convolutional layers and a neuron-specific readout layer, parameterized as a rank-1 decomposition of weights across x-pixels ($\mathbf{w}_x$), y-pixels ($\mathbf{w}_y$), and convolutional channels ($\mathbf{w}_c$). **c** Example neural activity and predictions on held-out test images. **d** Distribution of the fraction of explainable variance explained across all neurons (FEVE, see "Methods", $N = 14,504$). **e** Performance as a function of training images ($N = 6$ mice). Error bars represent standard error of the mean (s.e.m.).

**f** Performance as a function of model depth compared to the Sensorium model (green)[24], and compared to a linear-nonlinear (LN) model (dashed line) ($N = 6$ mice). Error bars represent s.e.m. **g** Example readout weights $\mathbf{w}_x$ and $\mathbf{w}_y$ as well as their combined spatial map $W_{xy}$. **h** Pooling diameter distribution, estimated from $\mathbf{w}_x$ and $\mathbf{w}_y$. **i** Natural and generated stimuli presented during neural recordings in monkey V1, figure from ref. 15. **j**–**n** Same as (**c**–**h**) for the our models fit to the monkey V1 dataset. **l** includes the baseline model from ref. 15 which has 5 layers. Error bars represent s.e.m.

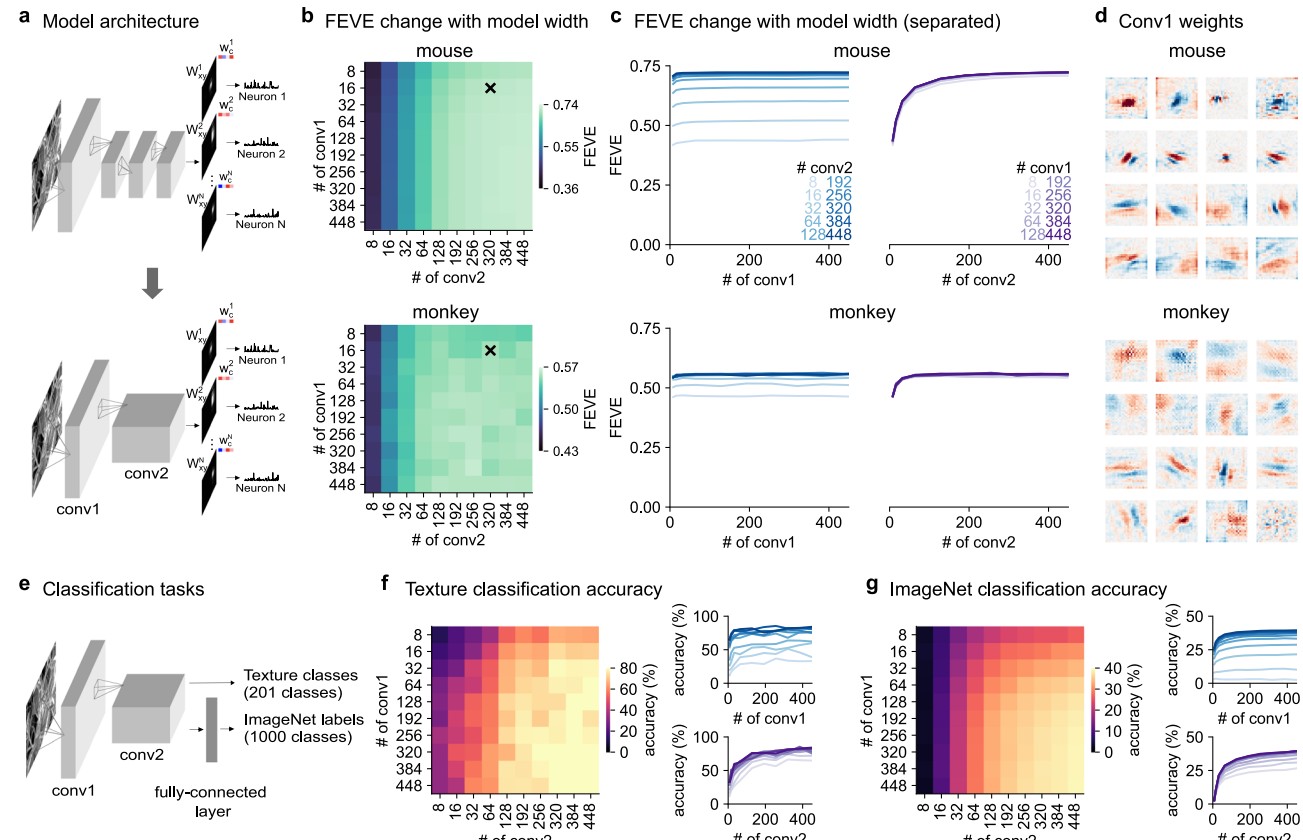

**Fig. 2 | Number of convolutional feature maps required to fit visual responses and perform visual tasks. a** Schematic of the simplified two-layer model. **b** Performance of the model as a function of the number of convolutional channels in the first layer (conv1) and the second layer (conv2) across 6 mice (top) and 2 monkeys (bottom). **c** Same data as (**b**) displayed as curves. **d** Conv1 weights for mouse (top) and monkey (bottom). **e** Two-layer convolutional models were trained to perform image classification on the ImageNet dataset or to perform texture classification on a dataset with 201 natural texture images. **f, g** Model performance as a function of the number of channels in conv1 and conv2 for texture classification and ImageNet classification.

diameter varied in the range of 10–25 degrees, consistent with the large receptive fields of mouse V1 (Fig. 1h[28,29]).

Next we repeated the same analysis for a publicly available dataset of monkey V1 neurons[15], consisting of 166 neurons recorded from two monkeys using silicon probes during the presentation of 7250 natural images (Fig. 1i). Similar to the mouse data, the FEVE saturated at two convolutional layers (Fig. 1l, 0.56 vs 0.56 FEVE for two vs four-layer models averaged across all neurons). This performance was higher than the best model from ref. 15 (0.50 FEVE), which required significantly more layers. The pooling weights, like in mice, were spatially restricted (Fig. 1m) and spanned a range of 0.15–0.8 degrees in diameter (Fig. 1n).

Thus, V1 models in both mouse and monkey required at most two hidden layers for near-maximal predictive power, resulting in a substantial simplification compared to previous high-performing models.

**Further simplifying the first layer to 16 convolutions**
To further simplify the two-layer model, we next varied the number of convolutional feature maps per layer (Fig. 2a). The performance did not decrease much when we made the first layer small, but did decrease substantially when we tried to make the second layer smaller (Fig. 2b, c). These results were qualitatively similar between mouse and monkey. Thus, we simplified both models to a first layer of 16 convolutions followed by a second layer of 320 convolutions, which we will refer to as the 16-320 model. Since the first layer convolutions are simply image filters, they can be visualized and interpreted easily (Fig. 2d, Supplementary Fig. 7a). In both mouse and monkey, the first

layer consisted of Gabor-like and center-surround filters with varying spatial frequencies and spatial extents. Removing the max pooling layer after the first layer resulted in smoother kernels but caused a slight decrease in performance (Supplementary Fig. 8).

The second layer consisted of a large number of channels, suggesting that a large expansion of dimensionality is needed to explain neural responses in the entire recorded populations. Indeed, when we applied a sparsity constraint to the readout layer $w_c$ weights[30], we found a drop in performance, demonstrating that the population model required a large number of conv2 channels (Supplementary Fig. 9a, b). We hypothesized that this architecture with expansion in the second layer may have computational advantages in visual tasks. To test this, we trained two-layer neural networks with different sizes to perform texture and object classification (Fig. 2e). For the texture classification task, we used random crops from 1001 large images of textures and trained the network to predict which image the crops were taken from, using a logistic regression readout (201 images were used for testing, see "Methods" for details). On this task, networks with wide second layers were necessary to achieve high accuracy (>200 convolutions, Fig. 2f). In contrast, the first layer provided good performance at 16 convolutional maps (Fig. 2f). We found a similar result when testing two-layer networks on an object recognition task based on a downsampled version of the ImageNet dataset (Fig. 2g)[31]. In this case, we had to use an intermediate fully-connected layer for good performance, as is typical for deep convolutional networks trained on ImageNet[32]. Two-layer networks trained on this task achieved up to 39% top-1 accuracy, comparable to the 43% accuracy of the deep

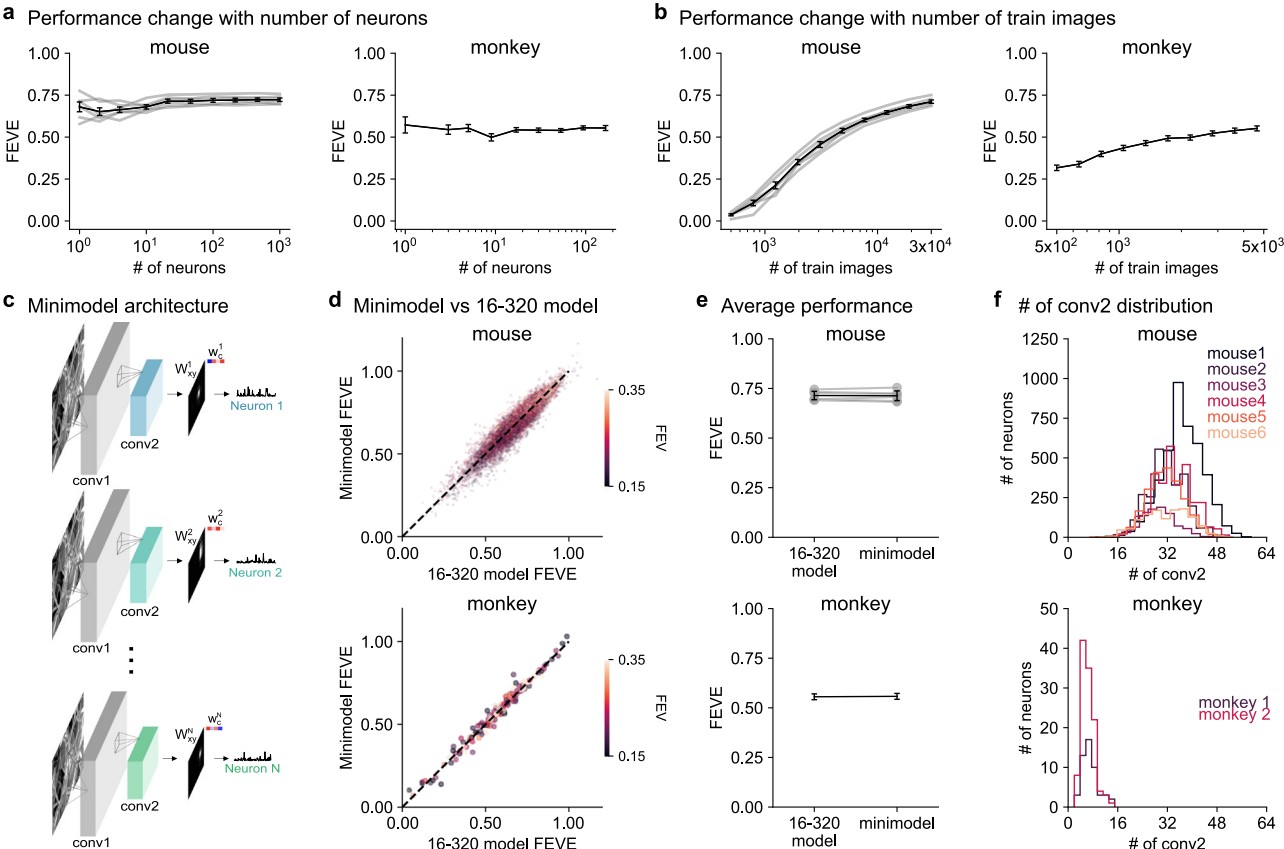

**Fig. 3 | Single neuron minimodels achieve similar performance to the 16-320 model. a, b** Performance of the 16-320 model does not improve with more neurons but improves with more training images. Error bars represent s.e.m. **c** Minimodel architecture consisting of a fixed conv1 layer from the 16-320 model, and separate conv2 and readout weights for each neuron. **d** Prediction performance of the minimodel compared to the 16-320 model for each neuron. **e** Same as (**d**) summarized per mouse/monkey. Error bars represent s.e.m. **f** Distribution of the number of conv2 feature maps in the mini-models.

Alexnet model with 5 hidden layers. Again we found that a wide second layer was necessary for good performance, while the first layer could be kept relatively small (Fig. 2g).

**Further simplifying the second layer to ~32 convolutions**
To further simplify the model, we wanted to reduce the size of the second layer. This was not possible when fitting all the neurons together as shown above (Fig. 2b, c), but it may become possible if the models are fit to individual neurons. This may be the case, for example, if the large conv2 dimensionality of the 16-320 model is only necessary because different neurons encode different small sets of features and thus a single 16-320 model would have to encode all those sets together. To test this hypothesis, we started by fitting the 16-320 model to groups of neurons of increasing number. We found little to no relation between the model performance and the number of neurons that were fit together (Fig. 3a). While population models showed a slight performance advantage compared to single-neuron models (Supplementary Fig. 10), this difference was minimal and disappeared when using the conv1 weights from a model trained on all neurons. Thus, there is no benefit in this analysis to recording and modeling thousands of neurons simultaneously. Indeed, the models fit to the monkey dataset performed very similarly. This was surprising to us, because the convolutional layer parameters are shared between neurons and thus could have benefited from the extra information provided by other neurons to reduce overfitting. In contrast, varying the number of training images had a large impact on performance, as expected from the extra information provided by additional training trials and as previously reported[26] (Fig. 3b).

Having found similar performance when fitting the 16-320 model separately to each neuron, we next tried reducing the second layer dimension for these single neuron models, which we will refer to as "minimodels" (Fig. 3c, Supplementary Fig. 11). Since we mainly wanted to investigate the effect of the second layer, we held the first layer fixed in all minimodels to the one identified by the 16-320 model (Fig. 2d). We did not directly vary the number of conv2 channels as done above (Fig. 2b, c), because that would have resulted in too many models to fit. Instead, we fit a single 16-64 model to each neuron with an added sparsity constraint on the readout layer[30]. The sparsity constraint pushed many readout weights to 0, thus effectively controlling the number of active convolutional maps. Every neuron ended with a different 16-X minimodel, where X was at most 64. The minimodels performed similarly to the 16-320 model, in both mouse and monkey neurons (Fig. 3d). On average, the minimodels achieved the same test set FEVE of 0.71 and 0.56 as the 16-320 models, respectively for the mouse and monkey models (Fig. 3e). The minimodels fit to the mouse data required an average of 32 convolutional maps in the second layer, while those fit to the monkey data required only 7 (Fig. 3f). There was no advantage in performance when using more maps in either model (Supplementary Fig. 9c, d). We also found no change in performance when using the conv1 filters from different mice to fit the minimodels (Supplementary Fig. 7b). Furthermore, there was no apparent clustering of conv2 weights or activities across mice when using the same conv1 filters (Supplementary Fig. 12). This suggests that each neuron's minimodel learns distinct features, collectively spanning a high-dimensional feature space.

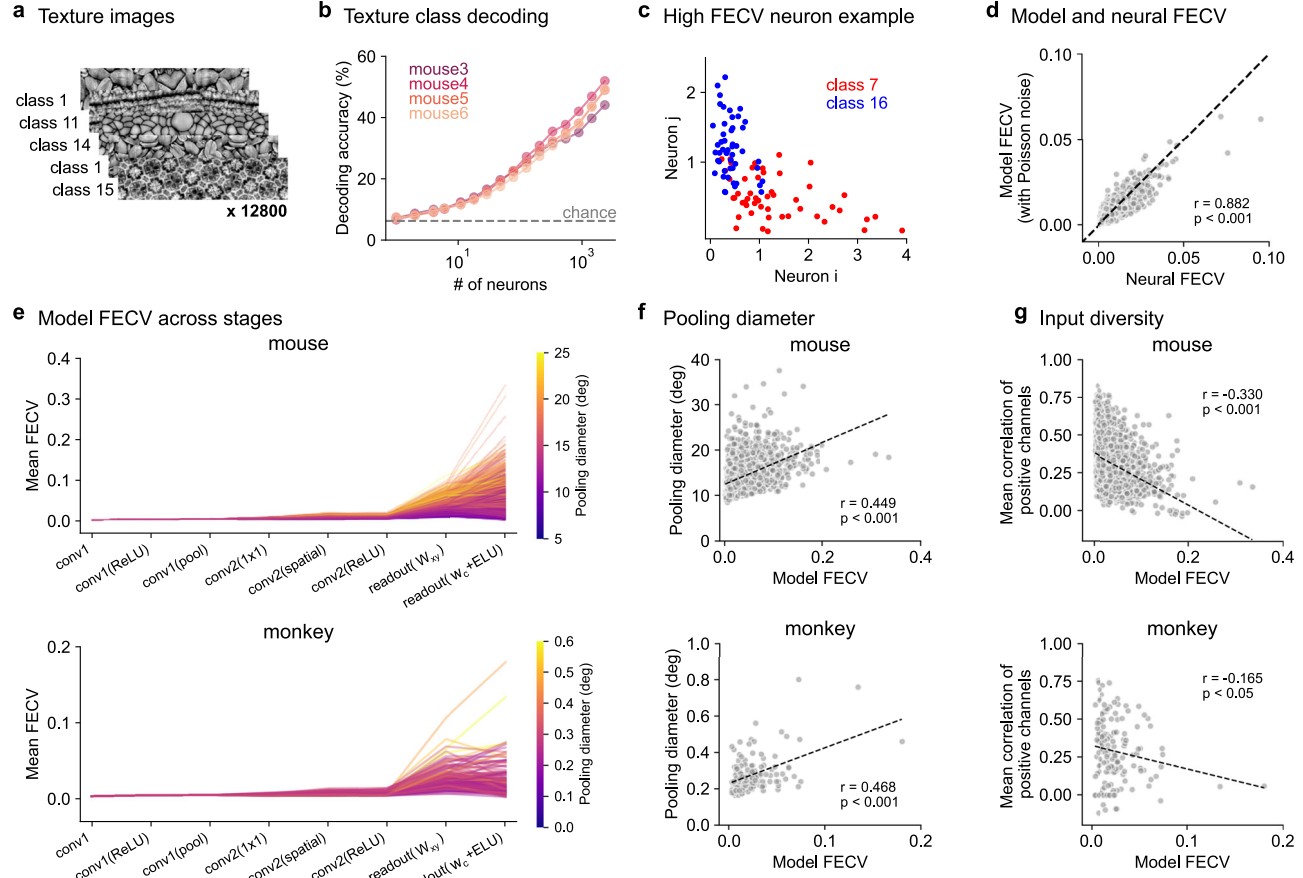

**Fig. 4 | Using minimodels to understand visual invariances. a** Visual textures from 16 categories were shown to the mice in addition to the natural images. **b** Decoding accuracy of texture class on test images ($n = 4$ mice). **c** Trial-averaged responses of two example neurons ($i$ and $j$) to the test images from two different texture classes (10 trials per test image). **d** Comparing the category variance of model neurons and recorded neurons. Pearson correlation ($r$) and $p$ value of two-sided test reported. **e** Mean category variance of model features after each successive operation. **f** Category variance of the model prediction vs pooling diameter in the readout layer. Pearson correlation ($r$) and $p$ value of two-sided test reported. **g** Same as (**f**) for the input diversity, which is defined as the mean correlation between conv2 channels with positive $\mathbf{w}_c$ weights. Pearson correlation ($r$) and $p$ value of two-sided test reported.

## Using minimodels to understand visual invariance

In this final section, we demonstrate the usefulness of the minimodels for understanding neural computations. We chose to look at neural invariances, which are thought to develop gradually in hierarchically organized neural systems, both artificial and biological. Using the minimodels, we can investigate whether the neural invariance indeed develops gradually, increasing at every stage in the model. Alternatively, some stages could result in a large jump in invariance while others may not contribute much. The invariance we investigated here was visual texture invariance, which may be relevant to both mouse and primate behavior[33–35]. We presented 16 classes of visual textures with 350 distinct exemplars in each class, created using random crops from 16 large photographs (Fig. 4a). The neural population represented well the visual texture category, which could be decoded with a single trial accuracy of 53.3% using logistic regression (Fig. 4b).

To understand the category encoding at the single-neuron level, we defined the fraction of explainable category variance (FECV) similarly to the FEV, but treating different exemplars from the same category as different "trials" in the FEV metric. We computed the FECV separately for all pairs of texture classes and then averaged the result across pairs (see "Methods"). Individual neurons with high pairwise category variance could indeed distinguish their respective categories well (Fig. 4c). The single-neuron category variance matched well the category variance computed from the respective

minimodels (Fig. 4d), suggesting that this property was captured well by the minimodels. Thus, we used the minimodels to try to understand how invariance arises in the V1 neurons. For both mouse and monkey neuron minimodels, we computed the FECV at different stages in the computation and found that the category variance primarily increased at the readout stage (Fig. 4e). This result stands in contrast with traditional views of hierarchical processing which predict a gradual increase in invariance at each layer of the hierarchy.

To further investigate how minimodel parameters influence texture invariance, we compared the FECV of the minimodels to the pooling diameter of the model readout and found a positive relationship (Fig. 4f). Thus, more pooling leads to more texture invariance. Finally, we hypothesized that the FECV could also depend on the diversity of features computed by the conv2 layer of each minimodel. We tested this using the mean correlation across positive conv2 channels as a measure of input diversity, and found that this measure was negatively correlated with the category variance (Fig. 4g). Thus, the less correlated the input channels are, the more likely it is for the model to have a high category variance.

Next, we visualized how category variance may arise in these neurons. As shown in (Fig. 5a), after spatial pooling with $W_{xy}$, each conv2 channel outputs a single feature activation for each stimulus. The neuron response can be interpreted as a weighted sum of these feature values, where the sign of the corresponding weight ($\mathbf{w}_c$)

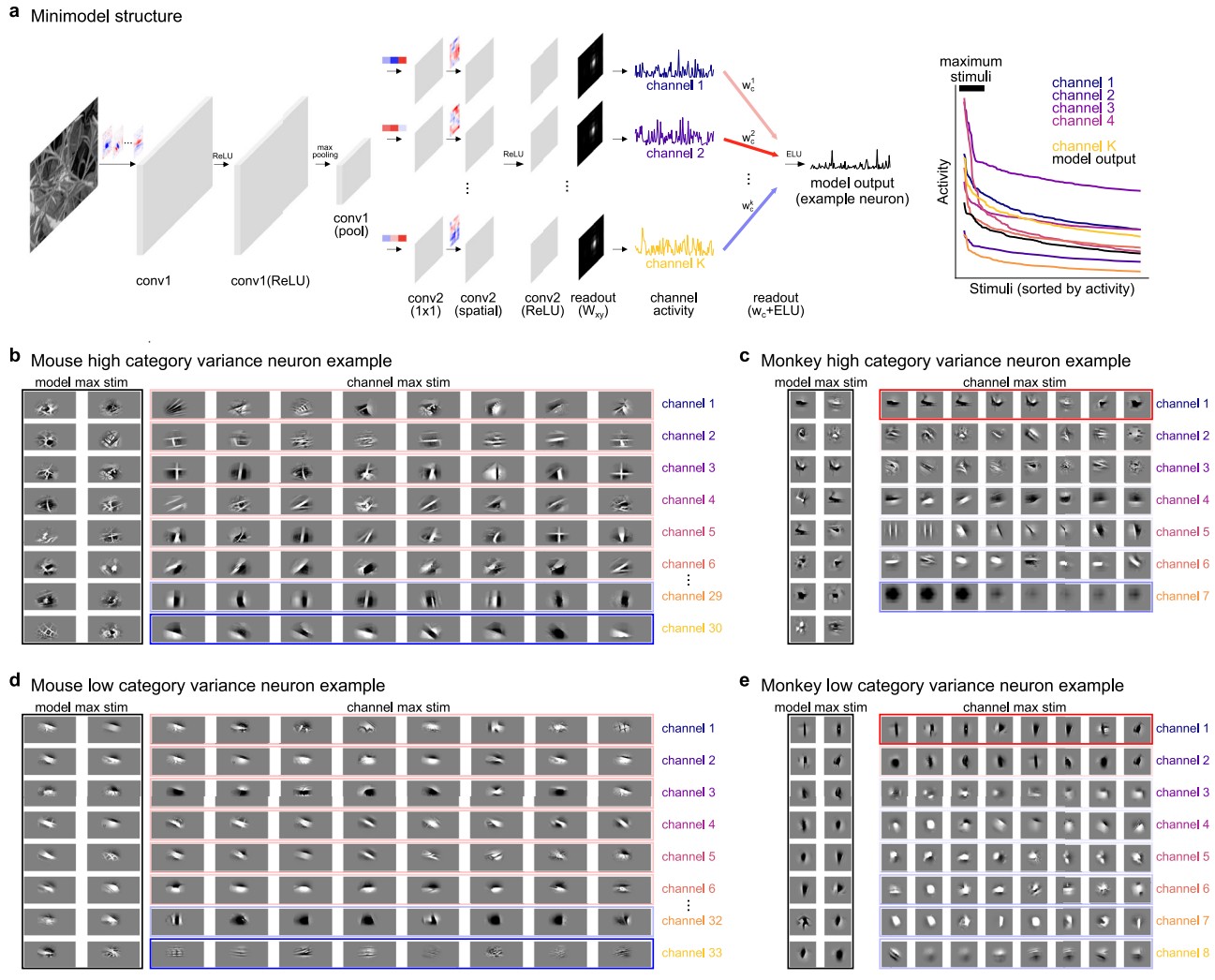

**Fig. 5 | Visualization of neurons using minimodels. a** Left: Schematic of the minimodel structure, including conv1 and conv2 layers, activities from each conv2 channel after pooling ($W_{xy}$), and their contributions to the predicted neural activity ($\mathbf{w}_c$). Right: Activity of each channel and the full model output, sorted. Maximum stimuli denoted by the black bar, are the top 16 stimuli. **b** Maximum stimuli for an example mouse neuron with high category variance (FECV). Left: Top 16 maximum stimuli for the full model output, masked by $W_{xy}$ (see "Methods"). Right: Top 8 maximum stimuli for the top 6 channels with the largest $w_c$ values and the bottom 2 channels with the smallest $w_c$ channels. The color and intensity indicates the $w_c$ value corresponding to each channel with red signifying positive weights and blue negative weights. **c**–**e** Same as (**b**) for other example neurons from mouse and monkey.

determines whether the channel contributes as excitatory or inhibitory input. For a subset of example neuron minimodels, we visualized the top 16 natural images which elicited the most activity (Fig. 5b–e, Supplementary Fig. 13). These image stimuli were more diverse for neurons with high category variance, as expected given these neurons are more texture invariant. We then visualized the top image stimuli for the conv2 channels for these neurons, and observed that in both high and low category variance neurons the top stimuli were less diverse than those from the minimodel output. This is also expected because we found a large increase in category variance from conv2 to the readout output across all neurons (Fig. 4e).

## Discussion

Here we have developed a class of simplified neural network minimodels that capture the response properties of mouse and monkey V1 neurons just as well as much larger and deeper neural networks (Supplementary Fig. 14). Using a new high-quality dataset of neural recordings, we showed that these models can explain a large fraction

of variance in neural responses (71%). The minimodels constitute a bridge between the complex and high-performing deep neural networks and simpler, more traditional models with limited predictive power but more potential for interpretation. We have also shown that minimodels can be used to understand properties of sensory computations such as invariance, and to formulate hypotheses about the neural stages at which invariance emerges. When viewed as a population of many minimodels, V1 appears to encode a large set of "layer 2" functions, built as linear combinations of "layer 1" filter responses. These properties imply that the classical filterbank view of V1 may need to be updated: V1 appears to represent a high-dimensional expansion of a low-dimensional filterbank. This updated model of V1 is substantially simpler than previous deep neural network models, and therefore we expect it to be more useful in generating testable predictions and useful descriptions of biological mechanisms. For example, we predict that the responses produced from "layer 1" in the model may resemble responses in LGN—past work supports this hypothesis as simple models can explain LGN responses[36–38].

Our results relate to previous descriptive models of V1, particularly models of complex cells. These models require combinations of "layer 1" filterbank responses to produce phase-invariant responses to edges and gratings[39]. However, the combinations required are typically simpler and more similar to the pooling stage of a convolutional layer, or to the pooling stage of the single-neuron readouts in our models. The dimensionality expansion we observed also relates to previous results showing that V1 neural responses are high-dimensional in both mice and monkeys[40,41]. The demonstration of texture invariance we have shown for mouse V1 neurons contrasts with results in primate V1, where the texture invariance in V1 appeared to be similar to the invariance properties of simple filterbanks[42]. The mechanism of generating invariance in the model (almost exclusively at the readout stage) also contrasts with the common view of a gradual build-up of invariance through multiple pooling and rectification operations[14,43]. From these findings, we predict that texture invariance is very low in the inputs to V1 neurons, and rather that this property emerges through the combination of diverse inputs to a single neuron. This view is consistent with previous studies that demonstrate the high computational capacity of a single neuron due to various forms of dendritic integration[44]. Testing this prediction may be possible with new glutamergic sensors which enable monitoring of many of the inputs to a single neuron simultaneously[45].

It remains to be seen whether higher-order visual areas require complex models to explain their responses—recent work suggests that smaller than expected models may work well in areas like primate V4[46]. Our results also suggest that the view of V1 as "complicated" may not necessarily apply to its sensory response properties. Previous results still imply that V1 neurons in mice are modulated by much more than just sensory inputs[47,48], and that V1 neurons can even modify their sensory tuning properties over the course of learning[49]. However, it remains to be seen whether these modulatory influences are in fact complicated, or perhaps similarly amenable to the kinds of simplifications we have shown here for sensory responses.

## Methods

All experimental procedures were conducted according to IACUC at HHMI Janelia. Data analysis and model fitting were performed in python using pytorch, scikit-learn and numpy, and figures were made using matplotlib and jupyter-notebooks[50–55].

### Data acquisition

**Animals.** All experimental procedures were conducted according to IACUC. We performed six recordings with many natural images and six retinotopic recordings in six mice bred to express GCaMP8s in excitatory neurons: TetO-jGCaMP8s x Camk2a-tTA mice (available as JAX 037717 and JAX 007004)[56]. These mice were male and female, and ranged from 2 to 12 months of age. Mice were housed in reverse light cycle, and were housed with siblings before and after surgery. Holding rooms are set to a temperature of $70\,°F \pm 2\,°F$, and humidity of $50\%rH \pm 20\%$.

**Surgical procedures.** Surgeries were performed in adult mice (P56-P200) following procedures outlined in ref. 8. In brief, mice were anesthetized with Isoflurane while a craniotomy was performed. Marcaine (no more than 8 mg/kg) was injected subcutaneously beneath the incision area, and warmed fluids +5% dextrose and Buprenorphine 0.1 mg/kg (systemic analgesic) were administered subcutaneously along with Dexamethasone 2 mg/kg via intramuscular route. For the visual cortical windows, measurements were taken to determine bregma-lambda distance and location of a 4 mm circular window over visual cortex, as far lateral and caudal as possible without compromising the stability of the implant. A 4 + 5 mm double window was placed into the craniotomy so that the 4mm window replaced the previously removed bone piece and the 5 mm window lay over the edge of the bone. After surgery, Ketoprofen 5 mg/kg was administered subcutaneously and the animal allowed to recover on heat. The mice were monitored for pain or distress and Ketoprofen 5 mg/kg was administered for 2 additional days following surgery.

**Imaging acquisition.** We used a custom-built 2-photon mesoscope[23] to record neural activity, and ScanImage[57] for data acquisition. We used a custom online Z-correction module (now in ScanImage), to correct for Z and XY drift online during the recording using the "MariusMotionEstimator" and the "MariusMotionCorrector". As described in ref. 8, we used an upgrade of the mesoscope that allowed us to approximately double the number of recorded neurons using temporal multiplexing[58], resulting in recordings at two depths simultaneously.

We first performed large field-of-view recordings (~3 Hz imaging rate) in each mouse in order to perform retinotopic mapping (Supplementary Fig. 1a, b). For the recordings with >30,000 natural images, the field-of-view was selected based on the retinotopic maps to ensure that neurons were in V1. The recordings used for analysis were performed at 30 Hz. In mice 1, 2, and 6, we imaged a larger area at two depths simultaneously (220 and 260 μm) using temporal multiplexing. In mice 3, 4, and 5, we imaged a smaller area at four total depths (100, 140, 220 and 260 μm). Each imaging session lasted two to 3 h.

During the recording, the mice were free to run on an air-floating ball. Mice were acclimatized to running on the ball for several sessions before imaging.

**Videography.** We used the same camera setup as in ref. 59. In brief, a Thorlabs M850L3 (850nm) infrared LED was pointed at the face of the mouse, and the videos were acquired at 50Hz using FLIR cameras with a zoom lens and an infrared filter (850 nm and 50 nm cutoff). The wavelength of 850 nm was chosen to avoid the 970 nm wavelength of the two-photon laser while remaining outside the visual detection range of the mice[60,61]. The camera acquisition software was a customized online version of Facemap[59].

**Visual stimuli.** We showed natural images on three tablet screens surrounding the mice (covering 270 degrees of the visual field of view). To prevent direct contamination of the PMT from the screen, we placed gel filters in front of the screen which exclude green light, and used only the blue and red channels. The original size of the natural images was 66 × 264 pixels, we cropped them to 66 × 130 pixels based on the horizontal retinotopy of the recording areas before training the model. Each pixel of the stimulus subtended ~1 degree of visual angle. To present the stimuli, we used PsychToolbox-3 in MATLAB[62]. The flashed visual stimuli were presented for 66.7 ms, alternating with a gray-screen image lasting 66.7 ms. Occasionally, the screen was left blank (gray screen) for a few seconds.

In all six mice, we presented a dataset comprised of a total of 32,440–52,868 natural texture images. Some of the images were horizontally flipped versions of each other. We selected 500 images from the dataset as the test set, and each of these images were presented up to ten times. The remaining images were used for training the model. The number of images in the training set was as follows for mice 1–6: 27,533, 33,753, 36,291, 33,955, 36,539, and 47,868. For each recording, 90% of the images in the training set were used for model training, while the remaining 10% were used for validation.

In four out of six mice, we presented a texture dataset consisting of 16 texture categories with 350 texture images per category. The texture images were randomly cropped from a high-resolution texture image from different locations, orientations and scales. These texture images were randomly presented together with the 30,000+ natural images. In each category, 300 texture images were used for training the classifier for decoding, 50 images are used for testing and each of the 50 test images are repeated 10 times. In order to look at higher-

level properties of texture coding, we removed low-level differences across image categories by matching their Fourier spectra on average. To do this, we calculated the average amplitude spectrum across all images in all categories, and then normalized the mean amplitude spectrum of each image category to this average. After that, we also adjusted the mean and standard deviation of the images from the 16 categories to match the mean and standard deviation of the other natural texture image dataset.

During image presentation, we performed online tracking of the left pupil of the mouse using a customized version of Facemap. When the eye position changed horizontally, we shifted the center-point of stimuli so that the stimuli were always presented as centered on the horizontal axis of rotation of the eye. To ensure that the stimulus appeared similarly to the mouse regardless of the eye rotation, we used a cylindrical projection of the monitor.

**Processing of calcium imaging data.** Calcium imaging data was processed using the Suite2p toolbox[63], available at www.github.com/MouseLand/suite2p. Suite2p performs motion correction, ROI detection, cell classification, neuropil correction, and spike deconvolution as described elsewhere[19]. For non-negative deconvolution (OASIS), we used a timescale of decay of 0.25 s[64,65]. We used the deconvolved traces for all analyses.

The temporal window used to calculate the stimulus response for each neuron consisted of two interpolated microscope frames, with the interpolation based on the time of the stimulus. The time delay for interpolation was selected to optimize the average fraction of explainable variance (FEV) for each recording, with 3.5 frames for mice 1 and 2, 4 frames for mice 3, 4 and 5, and 4.5 frames for mouse 6 (117 ms for mice 1 and 2, 133 ms for mice 3, 4, and 5, 150 ms for mouse 6). The stimulus responses were normalized for each neuron by division with the standard deviation, as done in previous studies[24,26].

**Monkey dataset.** We used a publicly available dataset[15], consisting of recordings from 166 V1 neurons in two monkeys using a linear 32-channel array spanning all cortical layers. The dataset contains 7250 images, each presented 1–4 times. We adopted the same train-test split as described in ref. 15, partitioning the data into training and testing sets, with 80% allocated for training and 20% for testing purposes. The dataset only contained neurons with an FEV > 0.15, so we did not filter the neurons in the recording.

Each image was displayed for 60 ms without gray screen intervals between images. Each image was masked by a circular aperture with a diameter of 2 degrees (140 pixels), featuring a soft fade-out effect starting at a diameter of 1 degree. Before fitting the model, the central 80 pixels (1.1 degrees) were cropped from the 140 pixels (2 degrees) square images.

## Retinotopy

Retinotopic maps for each imaging mouse were computed based on receptive field estimation using neural responses to natural images (at least 500 different natural images repeated three times each), as in ref. 33 (Supplementary Fig. 1a, b). In brief, this proceeded by (1) obtaining a well-fit convolutional model of neural responses with an optimized set of 200 spatial kernels, using a reference mouse; (2) fitting all neurons from our imaging mice to these kernels to identify the preferred kernel and the preferred spatial position; (3) aligning spatial position maps to a single map from the reference mouse; and (4) outlining brain regions in the reference mouse using spatial maps and approximately following the retinotopic maps from[66].

## Metrics

We used the same metrics to evaluate model performance in predicting neural responses as defined in refs. 15,24.

**Fraction of Explainable Variance (FEV).** Fraction of Explainable Variance (FEV) quantifies the proportion of the total variance in neuronal responses that can be attributed to the stimulus, excluding the noise variance. It is computed as the ratio between the explainable variance and the total variance. Specifically, the explainable variance is the total variance minus the variance of the observation noise. The FEV is used to select neurons for reporting model performance, and we include only those neurons with FEV greater than 0.15, following the methodology used in refs. 15,24.

The FEV is calculated using the following equation:

$$\text{FEV} = \frac{\text{Var}[r] - \sigma^2_{\text{noise}}}{\text{Var}[r]} \quad (1)$$

where Var[$r$] represents the total variance of the neural response $r$ across all stimuli and repetitions, and $\sigma^2_{\text{noise}}$ is the variance of the observation noise, estimated as the average variance across repetitions of the same stimulus:

$$\sigma^2_{\text{noise}} = \mathbb{E}_i[\text{Var}_j[r_{i,j}]]. \quad (2)$$

Here, $r_{i,j}$ is the response to the $j$-th repetition of the $i$-th image.

**Fraction of Explainable Variance Explained (FEVE).** Fraction of Explainable Variance Explained (FEVE) measures the proportion of the explainable variance that is captured by the model. It is defined as the ratio of the variance explained by the model to the explainable variance. This metric is critical for assessing how well the model accounts for stimulus-driven variations in neuronal responses.

The FEVE is calculated as follows:

$$\text{FEVE} = 1 - \frac{\frac{1}{N}\sum_{i,j}(r_{i,j} - \hat{o}_i)^2 - \sigma^2_{\text{noise}}}{\text{Var}[r] - \sigma^2_{\text{noise}}} \quad (3)$$

where $N$ is the total number of trials from $I$ images and $J$ repeats per image, $r_{i,j}$ is the observed response, $\hat{o}_i$ is the model's prediction for the $i$-th image, Var[$r$] is the total response variance, and $\sigma^2_{\text{noise}}$ is the observation noise variance.

## Population model of visual responses

Our full population model consists of a core, which is shared across all neurons, and a readout layer, which is distinct for each neuron, as in ref. 26 (Fig. 1b). The core has one to four layers, with each layer consisting of a convolutional layer (without a bias term), a batch normalization layer and an ELU nonlinear function. A 2 × 2 max pooling layer is applied after the first convolutional layer. The kernel size in the first layer is 25, the second layer is 9, and in each subsequent layer the kernel size is 5. Each convolution layer is depth separable other than the first layer, and all convolutional layers were initialized with Xavier initialization[67], as in ref. 26.

The readout layer is factorized using three rank-1 weight vectors: $\mathbf{w}_c$, $\mathbf{w}_x$, and $\mathbf{w}_y$. This readout is a simplified version of a previously proposed factorized model, in which the readout is divided into "where" and "what" components using a rank-2 weight matrix $W_{xy}$ and a rank-1 vector $\mathbf{w}_c$[27]. We further simplified the readout by enforcing non-negativity constraints on the $\mathbf{w}_x$ and $\mathbf{w}_y$ vectors, by clamping them above zero after each optimization step. The initial readout weights were drawn from a random normal distribution with 0.01 standard deviation.

In Fig. 1, we used 192 kernels per convolutional layer. In Fig. 2, we varied the number of kernels in each layer of the two convolutional layer model. In Fig. 3a, b, we used 16 and 320 kernels in the two convolutional layer model.

**Fitting procedure.** We used the Poisson loss between the neural responses and the model output as the cost function for training, which is a more biologically-plausible cost function for neural signals[26]. We used the AdamW[68] optimizer with a weight decay of 0.1 for the weights in the convolutional layers, 1.0 for $\mathbf{w}_x$ and $\mathbf{w}_y$, and 0.1 for $\mathbf{w}_c$. For the monkey data, overfitting occurs with the larger models, so we set the convolutional layer weight decay to 0.2 in the 3-layer model, and 0.3 in the 4-layer model. The training process consisted of four periods: the first period has 100 epochs (one epoch is a complete pass through all the training images) with a learning rate of 0.001, while the subsequent three periods each have 30 epochs and a learning rate reduced by a factor of 3 compared to the previous period. Finally, we selected the model across training epochs which demonstrated the best performance on the validation dataset.

For each data point in the analysis of performance change with the number of training images (Fig. 3a), we randomly sampled subsets of images from the training set while using the full validation image set for validation. For each data point in the analysis in Fig. 3b, we randomly sampled subsets of neurons using different random seeds. The number of seeds decreased logarithmically with the number of neurons. This approach aims to reduce the variability associated with smaller neuron subsets.

For the mouse datasets, we fit one model to each mouse and report both the average performance across all neurons (Fig. 1d) and the average performance across mice (Fig. 1e, f). For the monkey dataset, we trained a single model across all 166 neurons from two monkeys, and reported the average performance across all neurons (Fig. 1k, l).

## Model comparisons
**Sensorium model.** We compared our full population model with the model used as a baseline in the Sensorium competition[24], which was first proposed in ref. 26 (Fig. 1f). This model consists of a core and a readout layer. The core has 4 convolutional layers with 64 channels per layer. The readout parameterizes the readout location of each neuron as a learnable gaussian function, and during test, each neuron reads out from a single location based on the learned gaussian function. We trained the model with the code from https://github.com/sinzlab/sensorium/blob/main/notebooks/model_tutorial/1a_model_training_sensorium.ipynb, using default training settings with the eye position input disabled (because in our dataset we corrected the eye position online).

**VGG model.** We compared our monkey model with the previously reported best model on the dataset[15], which extracts visual features of the input image from the convolutional layers in the VGG-19 model and predicts the activity of each neuron with a generalized linear model (Fig. 1l). The VGG-19 model was pretrained on the ImageNet dataset[31,15] showed that the fifth convolutional layer (named "conv3_1") best predicts the neuron responses. The 5 convolutional layers have 64, 64, 128, 128 and 256 channels, with two pooling layers after the second and the fourth layer.

**Linear-nonlinear model.** We fit a LN model to each mouse dataset and to all monkey neurons (Fig. 1f, l). To create the LN baseline model, we removed the nonlinear activation functions from the core of a 16-320 model and replaced the max pooling layer with an average pooling layer. The LN baseline model was trained using the same procedure as the population models.

**Gabor model.** We use the same model as in ref. 40. We constructed 6720 Gabor filters, with parameters spatial frequency $f$ (0.1, 0.25, 0.5, 1, 2), orientation $\theta$ (0, pi/8, pi/4, 3*pi/8), phase $\psi$ (0, pi/4, pi/2, 3*pi/4, pi, 5*pi/4, 3*pi/2, 7*pi/4), size $\alpha$ (0.75, 1.25, 1.5, 2.5, 3.5, 4.5, 5.5), and eccentricity $\beta$ (1, 1.5, 2).

Simple cell responses were simulated by passing the dot product of the image with the filter through a rectifier function $r(x) = \max(0, x)$. Complex cell responses were simulated as the root-mean-square response of each unrectified simple cell filter and the same filter with phase $\psi$ shifted by 90°. The activity of a neuron was predicted as a linear combination of a simple cell and its complex cell counterpart, weighted by $C_1$ and $C_2$, which were estimated by linear regression. Each neuron was assigned to the filter which best predicted its responses to the training images and validation images (downsampled to $33 \times 65$ pixels). Neurons were classified as complex cells if the ratio between $C_2$ and $(C_1 + C_2)$ is larger than 0.5, and as simple cells if this ratio was less than or equal to 0.5.

This simple/complex Gabor model achieved 0.21 FEVE on the mouse dataset (with 14,504 neurons FEV > 0.15). We only keep the neurons with non-negative FEVE for the analysis in Supplementary Fig. 4b–e (12637/14504 neurons).

## Minimodels (per neuron models of visual responses)
We fit the minimodels separately to each single neuron, resulting in a distinct minimodel for each neuron (Fig. 3c). Each minimodel has two convolutional layers and a readout layer. The first layer consists of the 16 kernels from the first layer of the population model trained on all the neurons. A 2x2 max pooling layer is applied after the first convolutional layer. The second convolutional layer uses a simplified version of a depth separable convolution, which has one spatial convolutional layer and one $1 \times 1$ conv layer, initialized with 64 kernels (Supplementary Fig. 11b). We further simplified the minimodel by replacing the ELU in the core with a ReLU, so that the channel contribution to the final responses is entirely determined by the sign of Wc. The readout layer in the minimodel has the same structure as the readout layer in the population model.

**Fitting procedure.** Similar to our population model, the minimodel used a Poisson loss for training, with the AdamW[68] optimizer with a weight decay of 0.1 for the weights in the core, 1.0 for $\mathbf{w}_x$ and $\mathbf{w}_y$, and 0.2 for $\mathbf{w}_c$. The first convolutional layer was initialized with the first convolutional layer from the full model and fixed during training; we also used the first convolutional layer from the 16-320 models from other mice and achieved similar performance (Supplementary Fig. 7a). The initial readout weights were drawn from a random normal distribution with 0.01 standard deviation for $\mathbf{w}_x$ and $\mathbf{w}_y$, and 0.2 for $\mathbf{w}_c$. The training process consisted of four periods: the first period has 100 epochs with a learning rate of 0.001, while the subsequent three periods each have 30 epochs with the learning rate reduced by a factor of 3 compared to the previous period. Finally, we selected the model across training epochs which had the best performance on the validation dataset.

We initialized the second layer of the model with 64 conv2 kernels. To find the smallest number of channels required in conv2 without a performance decrease, we added a Hoyer-Square regularizer to the weights $W_c$. The Hoyer-Square regularizer is the square of the ratio between the $L1$ and $L2$ norms. The Hoyer-Square regularizer controls sparsity without reducing absolute weight values and is defined as follows[30]:

$$H_S(\mathbf{w}) = \frac{\left( \sum_i |w_i| \right)^2}{\sum_i w_i^2} \tag{4}$$

where $\mathbf{w}$ represents the weights of a layer, $w_i$ denotes the individual weights, $\sum_i |w_i|$ is the $L1$ norm, and $\sum_i w_i^2$ is the $L2$ norm of the weights. This regularization encourages sparsity in the $\mathbf{w}_c$ weights, enabling us to identify the minimal set of channels necessary for maintaining performance.

After training, we defined the number of the conv2 channels in each minimodel as the number of non-zero values in $\mathbf{w}_c$. To determine the strength of the sparsity loss, we randomly selected 10 neurons from the mouse 1 dataset and the monkey dataset, and selected the sparsity parameter that gives the smallest number of conv2 channels with a performance drop less than 1% on the validation set from the performance without sparsity penalty (Supplementary Fig. 9c). We showed that these sparsity penalties generalized well to other neurons, by computing the test performance and the number of conv2 channels obtained by fitting a different randomly selected set of 1000 neurons in the mouse dataset and by fitting all 166 neurons in the monkey dataset(Supplementary Fig. 9d).

## Model properties

**Pooling diameter.** The pooling diameter in Fig. 1 is computed using the $\mathbf{w}_x$ and $\mathbf{w}_y$ after smoothing with a gaussian of standard deviation of 3. Then the width at half-max of $\mathbf{w}_x$ and $\mathbf{w}_y$ is computed, and converted from pixels into degrees of visual angle. The pooling diameter is defined as the geometric mean of the two widths at half-max.

**Correlation of positive channels.** In the minimodels, the contribution of each channel in conv2 to the prediction is determined by the corresponding $\mathbf{w}_c$ value of each channel. The sign of the $\mathbf{w}_c$ indicates whether a certain channel contributes positively or negatively to the prediction. To measure how similar the features of each channel are, we only consider the excitatory channels, and calculate the correlation between each pair of channels based on the channel feature activations on the test images weighted by the $\mathbf{w}_x$ and $\mathbf{w}_y$ from the readout.

**Masked top stimuli.** To visualize the image features to which a neuron is most responsive, we plotted the stimuli which drove the most activity in the minimodel for the neuron, or which drove the most activity in the conv2 channels, and masked these stimuli based on the size and shape of the readout weights $\mathbf{w}_x$ and $\mathbf{w}_y$. The stimulus mask ellipse was centered at the maximum readout position in $x$ and $y$, with the initial width defined in $x$ and $y$ using the width at half-max for $\mathbf{w}_x$ and $\mathbf{w}_y$ respectively. The mask was then dilated by the size of the conv1 kernel in $x$ and $y$, which was 25 pixels, and each pixel in the mask was weighted by $\mathbf{w}_x$ and $\mathbf{w}_y$.

## Texture analysis

**Decoding accuracy.** We trained a 16-way logistic regression decoder with an L2 regularization strength of 10 on the neural responses to the texture training images in order to predict the texture categories. Prior to testing, we averaged the responses of the test images across ten repeats and Z-scored each neuron's activity using the mean and standard deviation calculated from the training images. We fit the decoder on random subsets of neurons and evaluated the classification accuracy on the test set (Fig. 4b).

**Fraction of category variance (FECV).** We define a new metric, the fraction of category variance, to measure the variability in a neuron's responses attributable to differences among categories after accounting for noise and accounting for signal variance within category.

Let $r_{ijk}$ represent the response of a neuron to the $j$-th repetition of the $i$-th stimulus in category $k$, and assume each category has $I$ images and $J$ repeats. The total variance $\sigma^2_{total}$ across all responses is empirically estimated as

$$\sigma^2_{total} = \frac{1}{N-1} \sum_{k=1}^{K} \sum_{i=1}^{I} \sum_{j=1}^{J} (r_{ijk} - \bar{r})^2 \tag{5}$$

where $\bar{r}$ is the overall mean response, and $N$ is the total number of responses.

The residual variance $\sigma^2_{residual}$, is the variance left after removing the category means, which removes the category variance:

$$\sigma^2_{residual} = \frac{1}{K} \sum_{k=1}^{K} \frac{1}{(IJ-1)} \sum_{i=1}^{I} \sum_{j=1}^{J} (r_{ijk} - \bar{r}_k)^2 \tag{6}$$

where $\bar{r}_k$ is the mean response to the category $k$.

The category variance $\sigma^2_{category}$ is calculated by subtracting the residual variance from the total variance:

$$\sigma^2_{category} = \sigma^2_{total} - \sigma^2_{residual}. \tag{7}$$

Finally, the FECV is calculated by dividing the $\sigma^2_{category}$ with $\sigma^2_{total}$:

$$FECV = \frac{\sigma^2_{category}}{\sigma^2_{total}}. \tag{8}$$

For the analysis of category variance, we included only the mouse minimodels with performance exceeding 0.7 FEVE, resulting in 3920 neurons being used for analysis. Similarly, we included only the monkey minimodels with performance exceeding 0.25 FEVE, resulting in 156 neurons being used for analysis.

To compare the FECV of the real neurons to those of the model neurons, we computed the activity of the model neurons in response to the texture stimuli, with Poisson noise added to match the neuronal noise (Fig. 4d). For each category, we have 300 images shown once and 50 images showing 10 times, resulting in 800 trials per category, and we use all trials to calculate FECV. We optimized the magnitude of the noise added in order to match the mean FEV of the model activity to the mean FEV of the real neurons within 0.1%. In Fig. 4e–g, we used the activations of each layer in the model neurons in response to the texture stimuli, without adding noise.

## Convolutional neural networks trained on tasks

The images for the texture classification task used 1001 high resolution images converted into grayscale and downsampled the images so that the minimum dimension was 256, then took a center crop to create images of size 256 by 256. We normalized the pixel intensities by the dataset mean of 128 and standard deviation of 61. The training augmentations were random rotation, random flipping, and random cropping into an image of size 112 by 112. The network was trained to predict which image the crops were taken from, using a cross-entropy loss. The images were randomly split into training and test images (800 and 201 respectively), and the network was first trained only with the training images. After training with the training images, the two convolutional layer weights were fixed and only the decoder part of the network was retrained on the test images to test the performance of the network core without overfitting. For retraining the decoder, three-quarters of each test image was used, with the bottom-right crop (128 by 128) of the image reserved for quantifying the accuracy of the network. On these held-out crops we predicted the class label from the network, and the accuracy was defined as the fraction of correct predictions (top-1 accuracy), with chance level at $\frac{1}{201}$.

The images for the ImageNet classification task were all the training and validation images from all 1000 classes in ImageNet, in RGB. We downsampled each image by a factor of 4 from their original size, and normalized by the overall channel means and standard deviations as is standard (mean = [0.485, 0.456, 0.406], std = [0.229, 0.224, 0.225])[31]. The training augmentations were random resized crop to size 64 by 64 with a scale range of 0.25–1.5 and random flipping. The network was trained to determine the image class for each crop out of 1000, using a cross-entropy loss. The validation images were never used for training, they were only used for determining the accuracy of the network. Each validation image was resized so that the minimum dimension was 74 and then a center crop from the resized image was

used of size 64 by 64. On these validation crops we predicted the class label from the trained network, and the accuracy was defined as the fraction of correct predictions (top-1 accuracy), with chance level at $\frac{1}{1000}$.

For both the texture discrimination and the image recognition (ImageNet) tasks, we trained networks with two convolutional layers, each followed by a batch norm layer and a ReLU nonlinearity. The convolutional filter sizes were 13 and 9 in the texture task and 11 and 5 in the ImageNet task, with a stride of 2 in the first layer. In the texture task, average pooling with a filter size of 3 was performed after the second convolutional layer. In the ImageNet task, max pooling with a filter size of 3 was performed after each of the convolutional layers, like in AlexNet[32]. In each task, the output of the last pooling layer was followed by a dropout layer, which was set to a dropout probability of 0.25 and 0.5 during training, for the texture and ImageNet classification tasks respectively.

A five convolutional layer network was also trained on the ImageNet task for comparison. This network had the same structure for the first two convolutional layers as above, then had three more convolutional layers each with a filter size of 3. The number of convolutional maps per layer were 64, 192, 384, 256, and 256.

The decoder for the texture task was a convolutional layer with a filter size of 1, which predicted the probability of each texture class. As in AlexNet, the decoder for the ImageNet task was a fully connected layer with a ReLU with size 4096, and a dropout probability of 0.5 during training, another fully connected layer with size 4096 and a ReLU, and finally another fully connected layer, which predicted the probability of each image class[32]. Default initialization was used for all weights, except for the first convolutional layer in the texture class network, which used the convolutional filters fit from one of the retinotopic mapping experiments as initialization.

The texture classification networks were trained with AdamW with a learning rate of 1e-3 and weight decay of 1e-5 and batch size of 16[68]. The full network was trained for 500 epochs if the second convolutional layer was smaller than 128 channels, and otherwise for 800 epochs. The decoder was then trained on the test images (excluding the bottom-right crops) for 300 epochs with the same learning rate and weight decay. We trained five networks with different random initializations at each conv1/conv2 map size and averaged the accuracies across these networks.

The ImageNet classification networks were trained with Adam for 180 epochs with an initial learning rate of 5e-4 and a batch size of 512[69]. The learning rate was annealed by a factor of 10 at epoch 100 and epoch 140.

## Reporting summary

Further information on research design is available in the Nature Portfolio Reporting Summary linked to this article.

## Data availability

We used a previously published monkey dataset of 166 neurons recorded in V1, available at https://doi.org.gin.g-node.org/10.12751/g-node. 2e31e3/[15]. The neural activity data generated in this study have been deposited in the Figshare repository at https://janelia.figshare.com/articles/dataset/Towards_a_simplified_model_of_primary_visual_cortex/28797638[21].

## Code availability

The code package for neural model fitting is available at https://github.com/MouseLand/minimodel/. The code to reproduce the figures is available at https://github.com/MouseLand/minimodel/tree/main/figures.

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

## Acknowledgements

This research was funded by the Howard Hughes Medical Institute at the Janelia Research Campus (F.D., M.N., M.P. and C.S.). We thank the GENIE project team, Caiying Guo, and Crystall Lopez at Janelia for generating the TetO-jGCaMP8s transgenic mice. We thank the Vivarium staff for animal husbandry, Sarah Lindo and Sal DiLisio for surgery support, Jon Arnold for designing headbars and coverslips, Dan Flickinger for microscopy support, and Jon Arnold and Tobias Goulet for engineering support.

## Author contributions

F.D., M.P. and C.S. designed the study. F.D. and M.N. performed data collection. F.D. performed data analysis. F.D., C.S. and M.P. wrote the manuscript with input from M.N.

## Competing interests

The authors declare no competing interests.
