## [Transparent Peer Review file · Nature Communications]

A simplified minimodel of visual cortical neurons

Corresponding Author: Dr Carsen Stringer

Version 0:

Reviewer comments:

Reviewer #1

(Remarks to the Author)

The premise of this paper is that complex, many-stage neural network models for visual cortical responses in mouse and monkey (which are predictive but not very interpretable) could be replaced by simpler few-stage models, which they will test by systematically removing components.

The authors convincingly show that much simpler models (2 layers, and in the simplest 'minimodel,' very small second convolutional layers) are as accurate at predicting neural responses from input images as more complicated (4-layer) models. With the very large new mouse V1 dataset, they also demonstrate dramatic model improvement when training sets have more samples; establishing the order of magnitude for how much data is needed to create an accurate model is an important result. Finally, using the simplified model, they examine texture category invariance by determining where in the network texture category is decodable. This invariance emerges at the readout layer of the second convolution, rather than gradually with each operation in the network.

My main concerns are around the significance and impact of the findings for neuroscience. It is a challenge to write an ML paper that neuroscientists will learn something from, and I think some more work could be done to bridge the communication gap between these fields.

Major:

1. It is interesting that responses in single neurons can be captured with relatively simple NN models, and that performance is robust to differences in the first layer across recordings. Is this simply a consequence of a universal approximation theorem, or is something more subtle happening?
2. What is the science insight that we get from this analysis and how does it drive the field forward? A line in the discussion notes that dendritic integration could perform these computations, which seems plausible, but not very specific. How would you test your model predictions? What new experiment/data do you need? The mouse recordings appear to be primarily in layer 2/3, if I understood the Methods correctly. What do you predict for layer 4? Or thalamic population activity?
3. Relatedly, the motivation for gradual emergence across a hierarchy is justified using recordings from cells from V2-V4, while the model here only accounts for individual neurons in V1. Is it surprising then that the emergence is sudden?
4. There are large differences between mouse V1 and monkey V1, and there are large differences in the datasets used (10^4 neurons in L2/3 of mouse V1; 166 neurons across all layers in monkey). It's not clear what to draw from the comparing the models fit here.

Minor comments:

1. What is the significance of spatially restricting pooling in the second convolutional layer? (lines 107-8) Was this enforced or did it emerge through fitting? If it's important, there should be some note in the discussion.
2. Watch out for jargon, especially where similar words have different meanings in ML vs neuro. Nat Comm is a journal with a broad audience and ideally the basic components of the multi-layer networks would have explicit descriptions in the Methods, rather than 'standard four-layer neural network model' (line 69) in the main text.

(Remarks on code availability)

I did not install and run the code.

I did note that in the checklist under 'data exclusion,' none was noted. However, the manuscript says that cells that were unresponsive (<15% fraction explained variance (FEV)) were dropped, as is standard. This should be reflected in the checklist.

Reviewer #2

(Remarks to the Author)

In this article, Du et al. proposed a class of simplified artificial neural network (ANN) models that could well predict response variances of neurons in the primary visual cortex (V1) of rodents and monkeys. They found that the proposed ANN models needed only two convolutional layers to perform well in predicting the response variances and performed significantly better than the classical linear-nonlinear model. The manuscript is clearly-written, and the data strongly support the authors' claims. Overall, the research topic would appeal to a broad range of visual neuroscientists. I have only a few comments that need to be sufficiently addressed.

1. The main advantages of the proposed ANN model were not made clear in the present manuscript. In Figure 1f, the FEVE value in the first convolution layer is significantly larger than in the Sensorium and linear-nonlinear models. Why is the proposed ANN model better at explaining the response variance than other models?

2. The authors found that the proposed model could well describe the receptive field properties of both mice and monkeys. However, there was a significant difference in the FEVE between Mice (mean = 0.73, figure 1d) and Monkeys (mean = 0.56, figure 1k). Is the FEVE difference due to different pooling sizes (figure 1h versus figure 1n) or other receptive field parameters such as preferred spatial frequency and linearity of spatial summation between the two species?

3. The range of the FEVE distribution is large and is probably related to the diverse receptive-field properties of V1 neurons. For example, the linear-nonlinear model could well predict the receptive field of simple cells ($F1/F0 > 1$) but not complex cells ($F1/F0 < 1$). Could the authors provide some hints regarding the relation between the amplitude of the FEVE and receptive-field properties such as simple/complex, preferred spatial frequency, etc.?

Minor: a few important details that I hope the authors could address

1. Figure 1 caption: (h) Pooling diameter distribution, estimated from w_x and w_y ($N=166$). The number of neurons in Figure 1h is incorrect ('166' is the number of neurons in the monkey dataset)
2. In Line 540: For clarification, I suggest adding the time in seconds to represent the optimal time delay instead of showing only the number of frames.
3. In supplementary figure 2: what is the reason for only showing monkey V1 neurons with $FEV > 15\%$ in S2C [all mouse V1 neurons are shown in S2a]?
4. In supplementary figures 3 and 5: could the authors provide a scale bar for each neuron so the reader can know the size of the receptive field?

(Remarks on code availability)

I do not have sufficient knowledge of installing and running the application. But I understand the concepts and the ideas the authors proposed in the manuscript.

Reviewer #3

(Remarks to the Author)

This is a nicely written, well-performed study of data-driven models of area V1. The authors' main claim, which is well-supported, is that neurons in mouse and macaque V1's responses to natural images can be explained by a relatively shallow CNN with 2 layers, the first being narrow, and the other being one wide. They show they can distill these models further for each neuron, providing compact descriptions of these neurons, which are potentially easier to interpret. They provide a normative account for why this might be the case, and demonstrate that their minimodels are useful for downstream investigations of invariance in area V1.

I enjoyed reading this manuscript. It is timely, in light of a recent preprint showing that V4 can also be explained by relatively compact networks (Cowley et al. 2023).

I have relatively minor concerns regarding the paper, that I think could be addressed in a revision. The first is that while the authors motivate their study in terms of finding models which are more compact and interpretable, they achieve their first goal without directly addressing the second goal. We don't see the model units of the second layer filters, their preferred stimuli, circuit analysis, etc. If the goal of making the model compact is to make it easier to interpret the network, this misses the mark—it is another black-box model with a large number of parameters, albeit with a smaller number of filters.

My other concerns center around two surprising findings, which are at odds with previous literature. They show that 2 layers are sufficient, whereas previous research has relied on deeper networks (e.g. Walker et al. 2019, Cadena et al. 2019, Lurz et al. 2021). While the authors show convincingly that two layers are enough, they don't show *why*. It could be that the discrepancy is simply due to the size of the receptive fields of the DNNs they use; they use unusually large filters compared to prior work. The study would be enhanced by determining which of the many hyperparameters and design choices that they've made drive this different result.

The second surprising finding is that there is no positive transfer from training neural networks on one vs. many neurons (Figure 3A). The authors themselves note that this is surprising, and indeed this is out of line with prior research. However, they don't provide an explanation for this phenomenon. Again, they should try and track down why this is different from prior literature.

Despite these minor concerns, I am confident that this paper will make a positive impact on the field and will be well

received, and I support the publication of this paper, with the minor reservations expressed above.

Below, I have collated detailed feedback.

Claims

- * They showed that neurons in mouse and macaque V1 responding to natural images can be explained by a 2-hidden-layer CNN
- * They showed that the optimal shape of the transformation is a small first layer and a wide second layer
- * They could make the second layer small without loss of performance, by distilling mini-models for each neuron
- * They found differences in the number of filters needed to explain mouse and monkey v1, with monkey v1 being explainable by a smaller number of second-layer filters
- * They performed an analysis of visual invariance in the context of texture discrimination and found that minimodels' invariance appeared at the very end of the processing, that is, at the readout layer

Positives

- * The paper is well-written, the figures are clear, and the methods are appropriate for the questions they investigate
- * They recorded a new large dataset containing 29,000 neurons responding to up to 65k natural image presentations in mouse V1 and are releasing it as open open-access
- * They did a thorough hyperparameter search that constrains the shape of the transformation in V1
- * They did a comparative study of mice vs. macaques, which is fairly rare
- * They show scaling laws, which allows one to tease apart effects of model form and amount of training data, which are often conflated in these studies
- * They give a hint at a normative explanation for why the V1 model has the shape it has in Figures 2e-g

Neutral

- * They try to demonstrate the usefulness of their minimodels for understanding neural computations. They do this by investigating neural invariances in a texture discrimination task. This might just be a matter of personal preferences and interests, but I didn't find this very relevant to the main story. The main story that they're trying to put forward, in my read of this paper, is that V1 responses can be explained by a relatively shallow network, which is narrow in its first layer and wide in the second layer, and that this is useful/normatively the right solution. If their claim is that minimodels are useful above and beyond existing larger-scale models, then figure 4 should focus on the fact that 1) minimodels are useful and 2) bigger models are not as useful. They show 1 but not 2. I suspect that other reviewers might have other preferences and interests here, so I'm not going to bat for this, but in my opinion, it takes space away from more analysis which is more germane to the main point they're putting forward.

Negatives

- * They start out with the premise that "this performance [of ANNs to explain V1] often comes at the expense of simplicity and interpretability". A shallower network is indeed "simpler" than a deeper network for some reasonable definition of simplicity—although I'd like a quantitative comparison in terms of number of weights compared to prior art. But interpretable? They don't engage in a lot of interpretation; we don't see the pattern of weights in the second layer, or their preferred stimuli. A wide layer with 384 filters feels quite big and unwieldy. Smaller second layers of up to 32 filters might be more interpretable, but they don't try to interpret these models either. It's entirely possible that the smaller models (especially ones with very small initial layers) are less sparse and less interpretable, similar to the phenomenon of superposition in transformers (Elhage et al. 2022, Toy models of superposition). In other words, I wouldn't conflate "compact" and "interpretable" as they seem to do.
- * Can the models be made more compact still? How similar are the second-layer filters to each other? I can imagine a world in which many of the second-layer filters they find are redundant, for instance having similar filters but different nonlinearities. See e.g. Martinelli et al. (2023) for work in that direction.
- * Their first layer filters look noisy and not very well estimated to me. They have checkerboard artifacts and are not very spatially localized. These filters are also unusually large at 25x25. Are the results robust to decreasing the size of the filters, which should improve their estimation?
- * What I'd like to see is whether they're the SAME layer 2 filters which are shared across the minimodels. They could then make a stronger claim that the minimodels are more interpretable than the base models. As is, the minimodels could have completely different layer 2 filters (layer 1 filters are frozen). I would be curious to see what happens when the readout sparseness penalty [the Hoyer Square penalty] is applied on the large model rather than on neuron-wise minimodels.
- * There are several surprising findings that are stated as fact but are not investigated further. Previous work showed that up to 4 layers are necessary to explain V1 neurons (e.g. Lurz et al. 2021), but they show that 2 is enough. Why? This is core to their claims, and as far as I can tell they don't provide a good mechanistic explanation why their results are so different than previous research. Lurz et al. (2021) don't list all the details of their models, which makes the comparison hard, but my read of the Sensorium competition sample code (https://github.com/sinzlab/sensorium/blob/main/notebooks/model_tutorial/0_baseline_CNN.ipynb) is that their filters are much smaller than here, at 13x13 for the first layer and 3x3 for the subsequent layers (please advise if a different reference model is used). Could it be that the discrepancy between the previously reported need for several layers is simply due to the size of RFs in the CNNs? That would be an interesting finding.
- * Another surprising finding they don't dig much into is the result in Figure 3a. This seems out of line with other studies. Lurz et al. (2021) found positive transfer of the core from one animal to another. At face value, the results of Du et al. (2024) seem

to be incompatible with these older results. What's going on here? Is it that they train on more images, obviating the need for transfer learning across neurons? I concur with the authors' assessment that this is surprising, and I would like them to dig deeper.

Minor issues

* Lines 664 to 668, I think the description of Figure 3a and 3b are swapped

* (optional) It could be visually indicated in Figure S6 using a snowflake icon that the first layer was frozen

(Remarks on code availability)

Version 1:

Reviewer comments:

Reviewer #1

(Remarks to the Author)

The revised manuscript is much clearer and a pleasure to read. The authors have adequately addressed my original concerns and corrected the minor issues in the original submission. Moving Figs S8/S9 to the main text improved the clarity of the texture-invariance section substantially.

No further revisions are needed. Recommend acceptance.

(Remarks on code availability)

Reviewer #2

(Remarks to the Author)

The authors have appropriately addressed all of my comments and modified the manuscript accordingly. I do not have any more questions.

(Remarks on code availability)

I do not have sufficient knowledge of installing and running the application. However, I understand the concepts and the ideas the authors proposed in the manuscript.

Reviewer #3

(Remarks to the Author)

As I stated in my initial review, I am a strong supporter of this publication, which elegantly shows, with well-thought-out methods, that the properties of neurons in V1 can be explained by relatively shallow mini-models, in both mouse and macaques.

My main (mild) criticism was around the interpretation of these mini-models. The authors have thoroughly addressed my criticism with additional analysis and visualization (Figures 5 and S13). This nicely connects the narrative they put forward in the initial sections of the paper around the simplicity of their models to their later explanation of texture sensitivity in Figure 4. This tidies up the narrative nicely.

A second point they've addressed was my question about the relative lack of advantage to scaling up (Figure S10). Indeed they did find a regime where scaling data helps, thus recapitulating and extending previous results. Finally, they also elegantly addressed the discrepancy between their results and previous state-of-the-art results, performing a series of ablations to track down where their better performance comes from (Figure S5 and S8).

In light of these significant improvements over the previous version—which I had enjoyed—I enthusiastically recommend publication.

(Remarks on code availability)

Reviewer #1 (Remarks to the Author):

The premise of this paper is that complex, many-stage neural network models for visual cortical responses in mouse and monkey (which are predictive but not very interpretable) could be replaced by simpler few-stage models, which they will test by systematically removing components.

The authors convincingly show that much simpler models (2 layers, and in the simplest 'minimodel,' very small second convolutional layers) are as accurate at predicting neural responses from input images as more complicated (4-layer) models. With the very large new mouse V1 dataset, they also demonstrate dramatic model improvement when training sets have more samples; establishing the order of magnitude for how much data is needed to create an accurate model is an important result. Finally, using the simplified model, they examine texture category invariance by determining where in the network texture category is decodable. This invariance emerges at the readout layer of the second convolution, rather than gradually with each operation in the network.

Thank you for the positive feedback.

My main concerns are around the significance and impact of the findings for neuroscience. It is a challenge to write an ML paper that neuroscientists will learn something from, and I think some more work could be done to bridge the communication gap between these fields.

Major:

1. It is interesting that responses in single neurons can be captured with relatively simple NN models, and that performance is robust to differences in the first layer across recordings. Is this simply a consequence of a universal approximation theorem, or is something more subtle happening?

Thanks for the question, we think this is not just a consequence of the universal approximation theorem, but the fact that inputs to V1 are similar across mice. The NN approximation theorem states that if you have enough data, and an infinite hidden layer, then you can approximate any function. We found in our analyses that we can continuously make the model simpler and simpler, with fewer and fewer units, and it still works. We think this is not a consequence of a universal approximation theorem, but rather the opposite, that this is evidence that this particular architecture corresponds to structure in the real mouse visual system which, in the case of the first layer, is similar across mice.

2. What is the science insight that we get from this analysis and how does it drive the field forward? A line in the discussion notes that dendritic integration could perform these computations, which seems plausible, but not very specific. How would you test your model predictions? What new experiment/data do you need? The mouse recordings appear to be primarily in layer 2/3, if I understood the Methods correctly. What do you predict for layer 4? Or thalamic population activity?

We have added to the discussion some studies in LGN – we hypothesize simpler models may apply there, but there are no studies with sufficiently many images presented to fully characterize layer 4 / LGN neurons in the way we were able to do for layer 2/3 neurons (Lines 358-362).

To test our model directly, we would need to quantify the inputs to the neuron – we think this could be possible through new sensors, such as glutamate sensors which enable recording of hundreds of inputs to a single neuron (Aggarwal et al 2023). This is a focus of future work and we have added this point to the discussion (Lines 389-391).

3. Relatedly, the motivation for gradual emergence across a hierarchy is justified using recordings from cells from V2-V4, while the model here only accounts for individual neurons in V1. Is it surprising then that the emergence is sudden?

Sorry, we did not use appropriate citations here. The difference between V2 and V4 reported in Okazawa et al is actually very small. We meant to cite modeling studies that interpret the entire primate ventral stream as a gradual extraction of abstract properties with increased invariance, as well as computational deep learning studies that show a gradual increase of invariance in convnets. It is on this background that our finding appears to be relatively surprising, even if it's just at the level of V1. It is possible that higher-order areas continue to increase this invariance, but in mouse there are not many opportunities to do so since there is no feedforward processing of information as it is believed to happen in the ventral stream. We changed the citations to reflect this.

4. There are large differences between mouse V1 and monkey V1, and there are large differences in the datasets used (10^4 neurons in L2/3 of mouse V1; 166 neurons across all layers in monkey). It's not clear what to draw from the comparing the models fit here.

Despite these differences, we have found a lot of similarities, which is evidence that these systems share more properties than commonly thought. For example, we found that both mouse and monkey V1 could be modeled with a shallow but wide network, and also that single neurons had diverse responses which could be captured by fitting small minimodels to each neuron separately. This second analysis does not require large population recordings, and thus the differences in the numbers of recorded neurons do not influence the conclusions. Using the same analysis on both datasets enabled us to find some differences between the mouse and monkey V1 neurons; for example, mouse V1 neurons pooled more second layer filters than monkey V1 neurons (Figure 3f).

Minor comments:

1. What is the significance of spatially restricting pooling in the second convolutional layer? (lines 107-8) Was this enforced or did it emerge through fitting? If it's important, there should be some note in the discussion.

Sorry for the confusion here, the spatially-restricted pooling is not enforced, it emerged during fitting - we have clarified this (Line 81). Neurons have spatially-restricted linear receptive fields, as have been described in Niell & Stryker 2008. Our results demonstrate that non-linear models also have spatially-restricted receptive fields, suggesting that neurons primarily receive inputs from neurons with similar receptive fields.

2. Watch out for jargon, especially where similar words have different meanings in ML vs neuro. Nat Comm is a journal with a broad audience and ideally the basic components of the multi-layer networks would have explicit descriptions in the Methods, rather than 'standard four-layer neural network model' (line 69) in the main text.

Thanks, we have no longer called it a standard four-layer model. The details on each layer in the model are in the Methods.

Reviewer #1 (Remarks on code availability):

I did not install and run the code.

I did note that in the checklist under 'data exclusion,' none was noted. However, the manuscript says that cells that were unresponsive (<15% fraction explained variance (FEV)) were dropped, as is standard. This should be reflected in the checklist.

Thanks, we have added this to the checklist.

Reviewer #2 (Remarks to the Author):

In this article, Du et al. proposed a class of simplified artificial neural network (ANN) models that could well predict response variances of neurons in the primary visual cortex (V1) of rodents and monkeys. They found that the proposed ANN models needed only two convolutional layers to perform well in predicting the response variances and performed significantly better than the classical linear-nonlinear model. The manuscript is clearly-written, and the data strongly support the authors' claims. Overall, the research topic would appeal to a broad range of visual neuroscientists. I have only a few comments that need to be sufficiently addressed.

Thank you for the positive feedback.

1. The main advantages of the proposed ANN model were not made clear in the present manuscript. In Figure 1f, the FEVE value in the first convolution layer is significantly larger than in the Sensorium and linear-nonlinear models. Why is the proposed ANN model better at explaining the response variance than other models?

We found that our model achieved a better FEVE with a shallower network than the Sensorium model because our readout contained spatial pooling (Wxy), whereas the Sensorium model is a point readout from a specific position (new Figure S5). We tested this by using the same core as the Sensorium model, and found that we still outperformed the Sensorium model, particularly in the case of a one-layer core. This suggests that the neurons pool features from a large area which cannot be accounted for using a point readout and a single conv layer. The Sensorium model did perform similarly to our model at three- and four-layers, because the model receptive field size increases with depth.

2. The authors found that the proposed model could well describe the receptive field properties of both mice and monkeys. However, there was a significant difference in the FEVE between Mice (mean = 0.73, figure 1d) and Monkeys (mean = 0.56, figure 1k). Is the FEVE difference due to different pooling sizes (figure 1h versus figure 1n) or other receptive field parameters such as preferred spatial frequency and linearity of spatial summation between the two species?

The FEVE difference, we suspect, is mainly from the number of images used to train the model. In Figure 1e, the mouse model trained with 5,000 images achieves 0.531 FEVE, which is comparable to the 0.56 FEVE in the monkey dataset (which contained 5,800 training images). Additionally, we show in Figure 3b that the monkey model performance increases with the number of training images, suggesting that more images would improve the fits to the monkey dataset.

3. The range of the FEVE distribution is large and is probably related to the diverse receptive-field properties of V1 neurons. For example, the linear-nonlinear model could well

predict the receptive field of simple cells ($F1/F0 > 1$) but not complex cells ($F1/F0 < 1$). Could the authors provide some hints regarding the relation between the amplitude of the FEVE and receptive-field properties such as simple/complex, preferred spatial frequency, etc.?

Thanks for the suggestion. We fit a Gabor model to each neuron which was a linear combination of a simple cell and complex cell response (new Figure S4). We found that there was no relationship between FEVE and the various parameters of the Gabor, including whether or not the cell was “complex” and its spatial frequency preference.

Minor: a few important details that I hope the authors could address

1. Figure 1 caption: (h) Pooling diameter distribution, estimated from w_x and w_y ($N=166$). The number of neurons in Figure 1h is incorrect ('166' is the number of neurons in the monkey dataset)

Thanks, we removed this.

2. In Line 540: For clarification, I suggest adding the time in seconds to represent the optimal time delay instead of showing only the number of frames.

Thanks, we have updated the text to include the time in seconds alongside the number of frames.

3. In supplementary figure 2: what is the reason for only showing monkey V1 neurons with $FEV > 15\%$ in S2C [all mouse V1 neurons are shown in S2a]?

The authors of the monkey V1 dataset only shared neurons with an $FEV > 0.15$ (thus we do not have the full distribution as in the mouse dataset). We have clarified this in the methods and the figure legend.

4. In supplementary figures 3 and 5: could the authors provide a scale bar for each neuron so the reader can know the size of the receptive field?

We added scale bars to these two figures.

Reviewer #2 (Remarks on code availability):

I do not have sufficient knowledge of installing and running the application. But I understand the concepts and the ideas the authors proposed in the manuscript.

Reviewer #3 (Remarks to the Author):

This is a nicely written, well-performed study of data-driven models of area V1. The authors' main claim, which is well-supported, is that neurons in mouse and macaque V1's responses to natural images can be explained by a relatively shallow CNN with 2 layers, the first being narrow, and the other being one wide. They show they can distill these models further for each neuron, providing compact descriptions of these neurons, which are potentially easier to interpret. They provide a normative account for why this might be the case, and demonstrate that their minimodels are useful for downstream investigations of invariance in area V1.

I enjoyed reading this manuscript. It is timely, in light of a recent preprint showing that V4 can also be explained by relatively compact networks (Cowley et al. 2023).

Thank you for the positive feedback.

I have relatively minor concerns regarding the paper, that I think could be addressed in a revision. The first is that while the authors motivate their study in terms of finding models which are more compact and interpretable, they achieve their first goal without directly addressing the second goal. We don't see the model units of the second layer filters, their preferred stimuli, circuit analysis, etc. If the goal of making the model compact is to make it easier to interpret the network, this misses the mark—it is another black-box model with a large number of parameters, albeit with a smaller number of filters.

Thanks for the suggestion, we have included visualizations of the second layer filter preferred stimuli from the minimodels in new main Figure 5. The model units in the second layer consist of a 9x9 spatial convolution and a 1x1 convolution summing the conv1 filters, we have added visualizations of these weights in new Figure S12.

My other concerns center around two surprising findings, which are at odds with previous literature. They show that 2 layers are sufficient, whereas previous research has relied on deeper networks (e.g. Walker et al. 2019, Cadena et al. 2019, Lurz et al. 2021). While the authors show convincingly that two layers are enough, they don't show *why*. It could be that the discrepancy is simply due to the size of the receptive fields of the DNNs they use; they use unusually large filters compared to prior work. The study would be enhanced by determining which of the many hyperparameters and design choices that they've made drive this different result.

Indeed, the receptive field is a main reason, but not because of the large filter size, but rather because of the spatial pooling in the readout layer. We show this finding in a new Figure S5. We note that the previous papers used smaller kernel sizes (size 9x9, Lurz et al 2021) but they also first downsampled their images by a factor of two - so our kernel sizes are in fact comparable. We found a small improvement from using the full-size image over the downsampled image (new Figure S8).

The second surprising finding is that there is no positive transfer from training neural networks on one vs. many neurons (Figure 3A). The authors themselves note that this is surprising, and indeed this is out of line with prior research. However, they don't provide an explanation for this phenomenon. Again, they should try and track down why this is different from prior literature.

In Lurz et al 2021, the authors found a modest increase in model performance when using 50 neurons vs 500 or 3500 neurons (Figure 3, oracle score of ~0.82 vs ~0.88). We also find a modest increase as a function of the number of neurons, shown in new Figure S10. This increase is larger as a percentage of baseline performance when we show fewer images (5,000), which is comparable to the previous studies.

Despite these minor concerns, I am confident that this paper will make a positive impact on the field and will be well received, and I support the publication of this paper, with the minor reservations expressed above.

Below, I have collated detailed feedback.

Claims

* They showed that neurons in mouse and macaque V1 responding to natural images can be explained by a 2-hidden-layer CNN

- * They showed that the optimal shape of the transformation is a small first layer and a wide second layer
- * They could make the second layer small without loss of performance, by distilling mini-models for each neuron
- * They found differences in the number of filters needed to explain mouse and monkey v1, with monkey v1 being explainable by a smaller number of second-layer filters
- * They performed an analysis of visual invariance in the context of texture discrimination and found that minimodels' invariance appeared at the very end of the processing, that is, at the readout layer

Positives

- * The paper is well-written, the figures are clear, and the methods are appropriate for the questions they investigate
- * They recorded a new large dataset containing 29,000 neurons responding to up to 65k natural image presentations in mouse V1 and are releasing it as open open-access
- * They did a thorough hyperparameter search that constrains the shape of the transformation in V1
- * They did a comparative study of mice vs. macaques, which is fairly rare
- * They show scaling laws, which allows one to tease apart effects of model form and amount of training data, which are often conflated in these studies
- * They give a hint at a normative explanation for why the V1 model has the shape it has in Figures 2e-g

Neutral

* They try to demonstrate the usefulness of their minimodels for understanding neural computations. They do this by investigating neural invariances in a texture discrimination task. This might just be a matter of personal preferences and interests, but I didn't find this very relevant to the main story. The main story that they're trying to put forward, in my read of this paper, is that V1 responses can be explained by a relatively shallow network, which is narrow in its first layer and wide in the second layer, and that this is useful/normatively the right solution. If their claim is that minimodels are useful above and beyond existing larger-scale models, then figure 4 should focus on the fact that 1) minimodels are useful and 2) bigger models are not as useful. They show 1 but not 2. I suspect that other reviewers might have other preferences and interests here, so I'm not going to bat for this, but in my opinion, it takes space away from more analysis which is more germane to the main point they're putting forward.

Thanks for pointing this out, this was not clear in the original manuscript. We find that single neurons in the larger model are combinations of many more conv2 filters (new Figure S9a-b), and the number of non-zero weights cannot be reduced without reducing model performance. Thus, the minimodels are simpler models of each neuron than the full model. These simpler models can be more easily visualized, e.g. new Figure 5, Figure S13.

Negatives

* They start out with the premise that “this performance [of ANNs to explain V1] often comes at the expense of simplicity and interpretability”. A shallower network is indeed “simpler” than a deeper network for some reasonable definition of simplicity—although I'd like a quantitative comparison in terms of number of weights compared to prior art. But interpretable? They don't engage in a lot of interpretation; we don't see the pattern of weights in the second layer, or their preferred stimuli. A wide layer with 384 filters feels quite big and unwieldy. Smaller second layers of up to 32 filters might be more interpretable, but they don't try to interpret these models either.

Our mouse minimodel has an average of 13,200 parameters in the two convolutional layers, most of which (10,000) are in the conv1 layer, which is fully interpretable. In comparison, the previous model (Lurz et al. 2021) has 39,680 parameters in its convolutional layers. As described above, we have added a new Figure S12 with visualizations of the conv2 weights, and visualizations of the conv2 channel maximum stimuli in new Figure 5. This shows that the minimodels can be visualized in a way that would be difficult or impossible in wider and deeper models.

It's entirely possible that the smaller models (especially ones with very small initial layers) are less sparse and less interpretable, similar to the phenomenon of superposition in transformers (Elhage et al. 2022, Toy models of superposition). In other words, I wouldn't conflate "compact" and "interpretable" as they seem to do.

As shown in the new plots (a-b) in Figure S9, we think that the 16-320 population model contains conv2 filters which are not well-aligned with the properties of the single neurons. When we apply a sparsity penalty on W_c in the model, the fitting performance declined substantially. This indicates that dense weights are necessary in the full model, which implies that the conv2 filters in this model are less directly related to the firing of each neuron.

With the minimodel approach, the user can choose a sparsity level and potentially explore using more weights if it appears that the weights are mixed in a non-interpretable way, which is not possible with the population model. For the minimodels, we chose a sparsity level which did not reduce performance (Figure S9c-d). We found at this sparsity level that neurons had varied levels of correlations across channels (Figure 4g), and that the correlation level was related to the texture invariance properties of the neuron. Non-zero correlations of conv2 channel responses suggest that the inputs to a neuron are correlated, which can be tested in vivo. Thus, creating a model without any “superposition” may not be biologically accurate.

Can the models be made more compact still? How similar are the second-layer filters to each other? I can imagine a world in which many of the second-layer filters they find are redundant, for instance having similar filters but different nonlinearities. See e.g. Martinelli et al. (2023) for work in that direction.

We tried to decrease the size of the minimodels by increasing the sparsity penalty (Hoyer loss), but this resulted in decreased performance (Figure S9c-d). We did not find clustering of the 1x1 conv2 filter weights across neurons (new Figure S12). However, we could not find the Martinelli et al 2023 paper so we were unsure what the reviewer had in mind.

* Their first layer filters look noisy and not very well estimated to me. They have checkerboard artifacts and are not very spatially localized. These filters are also unusually large at 25x25. Are the results robust to decreasing the size of the filters, which should improve their estimation?

As noted above, the filters are not large relative to past papers. We did not find large differences in performance with smaller kernel sizes (new Figure S5). We found that the rasterization was due to the max pooling after the first layer - if we remove the max pool and reduce the image size instead, the conv filters are more smooth but performance is reduced (new Figure S8).

* What I'd like to see is whether they're the SAME layer 2 filters which are shared across the minimodels. They could then make a stronger claim that the minimodels are more interpretable than the base models. As is, the minimodels could have completely different layer 2 filters (layer 1 filters are frozen). I would be curious to see what happens when the readout sparseness penalty [the Hoyer Square penalty] is applied on the large model rather than on neuron-wise minimodels.

Thank you for raising this point. We trained minimodels with the same conv1 weights and examined whether layer 2 filters and channel features clustered across neurons. As shown in the new Figure S12, there was no clear clustering of layer 2 filters or channel features, indicating that these filters are learned uniquely for each neuron. This suggests that the minimodels adapt to neuron-specific features, which enhances their interpretability compared to population models.

We also applied the Hoyer-Square sparseness penalty directly to the population model (Figure S9, new plot a-b). As the penalty strength increased, we observed a drop in performance, further supporting the idea that population models require a larger number of shared features to account for the diverse feature sets across neurons.

* There are several surprising findings that are stated as fact but are not investigated further. Previous work showed that up to 4 layers are necessary to explain V1 neurons (e.g. Lurz et al. 2021), but they show that 2 is enough. Why? This is core to their claims, and as far as I can tell they don't provide a good mechanistic explanation why their results are so different than previous research. Lurz et al. (2021) don't list all the details of their models, which makes the comparison hard, but my read of the Sensorium competition sample code (https://github.com/sinzlab/sensorium/blob/main/notebooks/model_tutorial/0_baseline_CNN.ipynb) is that their filters are much smaller than here, at 13x13 for the first layer and 3x3 for the subsequent layers (please advise if a different reference model is used). Could it be that the discrepancy between the previously reported need for several layers is simply due to the size of RFs in the CNNs? That would be an interesting finding.

As described above, our model achieved a better FEVE with a shallower network due to the spatial pooling learned in the readout layer for each neuron, as shown in new Figure S5. We also did not find a large dependence on filter size (new Figure S5 and S8).

* Another surprising finding they don't dig much into is the result in Figure 3a. This seems out of line with other studies. Lurz et al. (2021) found positive transfer of the core from one animal to another. At face value, the results of Du et al. (2024) seem to be incompatible with these older results. What's going on here? Is it that they train on more images, obviating the need for transfer learning across neurons? I concur with the authors' assessment that this is surprising, and I would like them to dig deeper.

Thanks for asking for clarification here, see our response above and new Figure S10.

Minor issues

* Lines 664 to 668, I think the description of Figure 3a and 3b are swapped

Thanks we clarified by swapping the text in the legend.

* (optional) It could be visually indicated in Figure S6 using a snowflake icon that the first layer was frozen

Reviewer #1 (Remarks to the Author):

The revised manuscript is much clearer and a pleasure to read. The authors have adequately addressed my original concerns and corrected the minor issues in the original submission. Moving Figs S8/S9 to the main text improved the clarity of the texture-invariance section substantially.

No further revisions are needed. Recommend acceptance.

Thank you!

Reviewer #2 (Remarks to the Author):

The authors have appropriately addressed all of my comments and modified the manuscript accordingly. I do not have any more questions.

Thank you!

Reviewer #2 (Remarks on code availability):

I do not have sufficient knowledge of installing and running the application. However, I understand the concepts and the ideas the authors proposed in the manuscript.

Reviewer #3 (Remarks to the Author):

As I stated in my initial review, I am a strong supporter of this publication, which elegantly shows, with well-thought-out methods, that the properties of neurons in V1 can be explained by relatively shallow mini-models, in both mouse and macaques.

My main (mild) criticism was around the interpretation of these mini-models. The authors have thoroughly addressed my criticism with additional analysis and visualization (Figures 5 and S13). This nicely connects the narrative they put forward in the initial sections of the paper around the simplicity of their models to their later explanation of texture sensitivity in Figure 4. This tidies up the narrative nicely.

A second point they've addressed was my question about the relative lack of advantage to scaling up (Figure S10). Indeed they did find a regime where scaling data helps, thus recapitulating and extending previous results. Finally, they also elegantly addressed the discrepancy between their results and previous state-of-the-art results, performing a series of ablations to track down where their better performance comes from (Figure S5 and S8).

In light of these significant improvements over the previous version—which I had enjoyed—I enthusiastically recommend publication.

Thank you!